# Direct genome-wide identification of G-quadruplex structures by whole-genome resequencing

Jing Tu [1✉], Mengqin Duan [1], Wenli Liu[1], Na Lu [1], Yue Zhou[1], Xiao Sun[1] & Zuhong Lu [1✉]

We present a user-friendly and transferable genome-wide DNA G-quadruplex (G4) profiling method that identifies G4 structures from ordinary whole-genome resequencing data by seizing the slight fluctuation of sequencing quality. In the human genome, 736,689 G4 structures were identified, of which 45.9% of all predicted canonical G4-forming sequences were characterized. Over 89% of the detected canonical G4s were also identified by combining polymerase stop assays with next-generation sequencing. Testing using public datasets of 6 species demonstrated that the present method is widely applicable. The detection rates of predicted canonical quadruplexes ranged from 32% to 58%. Because single nucleotide variations (SNVs) influence the formation of G4 structures and have individual differences, the given method is available to identify and characterize G4s genome-wide for specific individuals.

---

[1] State Key Laboratory of Bioelectronics, School of Biological Science and Medical Engineering, Southeast University, 210096 Nanjing, China.
✉email: jtu@seu.edu.cn; zhlu@seu.edu.cn

The G-quadruplex (G4) structure is an alternative conformation of the double-helical structure of B-form DNA that arises in guanine-rich sequences[1]. G4s have also been discovered in single-stranded RNA sequences[2,3]. One four-stranded G4 is a stack of two or more planar G-tetrads, and one G-tetrad forms through Hoogsteen hydrogen bonding of four guanines[4]. G4 structures are considered to be knot-like structures and have been shown to form stably under near-physiological conditions in vitro[5,6]. G4 structures are stabilized by cations with a stabilizing preference of monovalent cations according to the following order: $K^+ > Na^+ > Li^+$[7]. The formation of G4 structures in the genome can influence chromatin architecture and many fundamental biological processes, such as DNA replication and gene regulation[8–10]. Identifying G4s of the whole genome or whole transcriptome will accelerate the study of the formation, stability, and function of these structures.

In the past decade, several methods have been developed to identify G4 in the whole genome or whole transcriptome, including computational sequence analyses and experiments. By searching simple consensus sequences[11,12] or accommodating structural variants[13–15], computational tools have helped to identify potential G4s in a given genome[16,17]. Computational predictions employ algorithms derived from experimental data on a small number of sequences and require thorough experimental validation. Using chromatin immunoprecipitation followed by high-throughput sequencing (G4 ChIP-seq), DNA G4s can be detected and mapped to the genome[18,19]. Antibodies against known G4-binding proteins are used to infer G4s in these approaches. Notably, G4 ChIP-seq depends on specific antibodies, and potential biases may be introduced[20,21]. The presence of folded G4 in the DNA can stall a DNA polymerase[22,23]. Coupled with high-throughput sequencing (G4-seq), more than 700,000 DNA G4 sites were identified based on a genome-wide DNA polymerase stop assay[24]. In G4-seq, each DNA template is sequenced twice. The initial sequencing run uses the monovalent cation $Na^+$ that does not stabilize G4s to ensure accurate sequencing, and the second sequencing run is under G4-stabilizing conditions (in $K^+$ or with the addition of pyridostatin (PDS)). By mismatch quantification based on the results of the first sequencing run, the folded G4s were detected genome-wide in vitro. Similarly, RNA G4 structures were detected in high-throughput using reverse transcriptase stalling (rG4-seq)[25]. G4-seq is a fast and powerful approach to identify G4 sites in the genome, which includes various noncanonical G4 structures that are difficult to predict. The optimized version of G4-seq has been used to generate whole-genome G4 maps for 12 species[26]. However, G4-seq depends on the accommodation of sequencing buffer, which is unreachable to most researchers. Two sequencing runs also mean double cost. The procedure will be more user-friendly and cost-effective if G4 structures can be directly profiled from standard high-throughput sequencing data.

Here, we present G4-miner, a genome-wide approach to directly identify G4 structures from standard sequencing data. G4-miner inspects sequencing quality in the whole genome, seizes unexpected fluctuations locally, and ascribes some of the quality fluctuations to the unstable formation of G4. We sequenced DNA from primary human B lymphocytes under standard Illumina sequencing conditions. Although the standard Illumina sequencing buffers do not cause strong G4 stabilization[24,27], a slight and inconstant drop in sequencing quality was recorded from the beginning of the G4 structure (Fig. 1). We MG4 sites in the human genome, appraised the results with the results of computational prediction and optimized G4-seq, and evaluated the influence of single-nucleotide variations (SNVs) in G4 formation using the seized G4 sites. The given method provides a user-friendly and transferable way to identify G4 structures genome-wide.

## Results

**Principle and process of G4-miner**. In G4-miner, DNAs are whole genome sequenced only once using a standard Illumina sequencing protocol and standard buffers. The standard sequencing buffers have been optimized and do not cause strong G4 stabilization[24,27]. However, destabilized G4 structures still influence the sequencing reaction. Instead of altering the sequencing readout, a slight and inconstant drop in sequencing quality was recorded from the beginning of the G4 structure (Fig. 1). In a standard sequencing experiment, DNA fragments in clusters were sequenced, and the Phred quality scores were calculated for each read (cluster). An unstably formed G4 causes unexpected fluctuations in sequencing quality, but usually does not cause mismatches.

G4-miner inspects sequencing quality in the whole genome, seizes unexpected fluctuations locally, and ascribes some of the quality fluctuations to the unstable formation of G4. Median quality scores of bases mapped to each locus in the genome were calculated after read mapping. For each site, we seize the quality scores of the next 75 nt and recorded the number of loci whose median quality score significantly dropped in comparison with that of the specific site. G4 sites were mined after setting the thresholds of the number of loci, and the quality score dropped.

**Validation by known sequences**. We validated our approach by inspecting the sequencing quality scores of four known control sequences: two containing stable G4 structures (src and myc), one conforming requirement of canonical G4, but that has been demonstrated to prevent G4 formation (a-repeat)[28], and a guanine-rich strand that cannot form a G4 (G-rich)[29] (Supplementary Table 1). Phred quality scores[30] were used to quantify the sequencing quality. We calculated the median quality scores for each locus in the template/genome to seize and represent inconstant sequencing quality fluctuations. Although the readouts were not altered in all control sequences, the median quality score fluctuated slightly from or before the beginning of the G4 structure in the positive controls (Fig. 2a, b). For src and myc, the median quality score dropped ≥ 4 at 55 and 25 loci among the 75 bases (half of the read length) from the beginning of the G4 structure. In contrast, sequencing of the negative controls observed fewer than 6 loci (Fig. 2c, d).

The inspection of quality scores and readouts along the positive control revealed that sequencing accuracy was slightly reduced after the G4 start sites, but the readouts were not altered (Fig. 1). At the same time, the reduction of quality scores was not observed in all reads that covered the G4 structure and the sequence downstream. Some of the reads had constant high-quality scores. The observations suggested that using standard Illumina sequencing buffers, the formation of G4s is not stable, and polymerase stalling does not occur in most template molecules. The sequencing quality score of a read was generated by comprehensively evaluating the accuracy of synthesis reactions in tens of thousands of molecules in a cluster. Under G4-destabilizing conditions (standard Illumina sequencing buffers), quadruplex structures were unstably formed in only a small portion of template molecules in a cluster. The pause of polymerase stalling that occurs in these molecules causes phase shift signals, which interfere with the real signals and induce a drop in the quality score.

**Whole genome seizing the quality fluctuation**. Next, we explored whether quality fluctuation could be specifically seized genome-wide. A set of 75 nt (half of the read length and longer than 99% predicted G-quadruplexes (PG4s); Supplementary Fig. 1) segments that contained ~178,606 PG4s and downstream

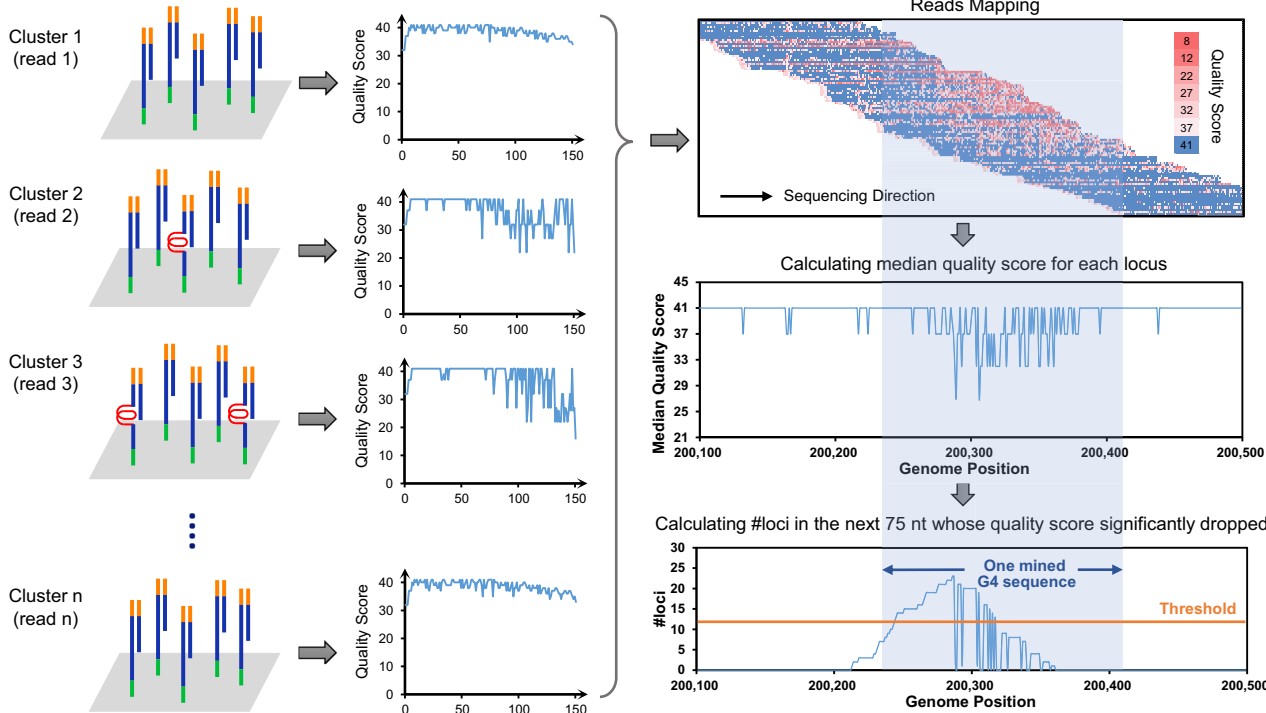

**Fig. 1 A schematic of the G4-miner method.** In a standard sequencing experiment, DNA fragments in clusters were sequenced, and the Phred quality scores were calculated for each read (cluster, orange and green parts are sequencing primers or adaptors, blue parts are the sequences to be determined, and the bent red parts are the sequences, which form G4 structures). The unstable formation of G4 causes unexpected fluctuations in sequencing quality, but does not cause mismatches. Read mapping was performed to calculate the median quality score of bases mapped to each locus in the genome. For each site in the genome, the number of loci in the next 75 nt whose median quality score significantly dropped in comparison with that of the specific site was calculated. G4 sequences in the whole genome can be mined after setting the threshold of the number of loci and the quality score dropped.

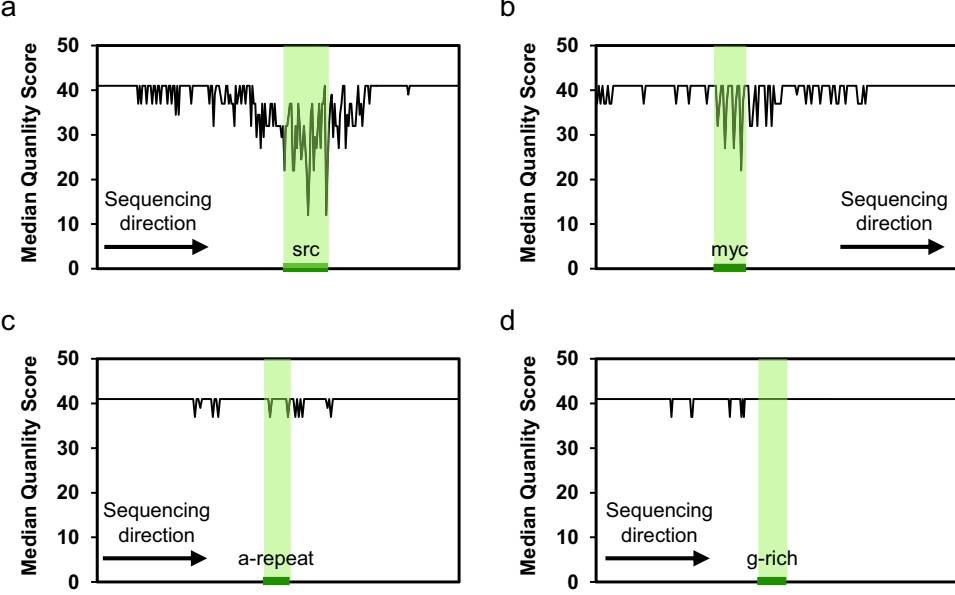

**Fig. 2 Sequencing quality of four known controls. a** Src, a sequence containing a stable G4 structure; **b** myc, a sequence containing a stable G4 structure; **c** a-repeat, negative control; **d** g-rich, negative control. The green shadows present the regions of the sequences.

sequences at the 3′ end were cut from the plus strand of the human genome. The remaining regions of the plus stand that did not contain PG4s were cut into 75 nt segments. The analysis based on approximately one million reads (~45× of the plus strand) generated by the standard pipeline showed more significant quality fluctuations in the segments containing PG4s

than the remaining segments. Over 40% of PG4-positive segments contained over 15 loci whose quality scores decreased no <4 in comparison with the loci before PG4s, and only 4.05% of non-PG4 segments passed the criteria (Supplementary Table 2). Considering that the guanine content is >28% in over 99% PG4-positive segments (Supplementary Fig. 2), we filtered the non-PG4

segments with low guanine content (G% <28%), and <1% of the non-PG4 segments remained under the same criteria (Supplementary Table 3). Although the sequence quality for most non-PG4s was stable with no observable decrease, the remaining 1% of non-PG4 segments were found to have significant quality fluctuation (~370,000 segments). The number of these non-PG4 fragments showed a considerable decline in a considerable number of loci, much greater than the predicted number of PG4 (178,606), which is consistent with the previous approach in G4-stabilizing conditions[24], indicating that the human genome contains more G4s than predicted[11].

We applied G4-miner to profile DNA G4 structures in the human genome by resequencing primary B lymphocytes (GM12878, Coriell Institute) using the Illumina HiSeq4000 platform under standard conditions. To identify reliable G4 sequences, we inspected the quality scores of 75 bases at the 3′ end of a specific site and screened median quality scores of each site in both strands of the human genome. We set individual thresholds for the two strands of the parallel runs to ensure that the false-positive rate was similar and smaller than 1% (Supplementary Table 4). The influence evaluation of sequencing depth revealed that the PG4 detection rate plateaued when the sequencing depth was greater than 20-fold for one stand (Supplementary Fig. 3). Therefore, two duplicates of the experiment were performed, and at least 1.88 billion reads were generated with an average coverage of 45× for each strand of the human genome (Supplementary Table 5). We set the threshold of 15 loci quality score decreased 4 (15, 4) for the plus strand of Experiment 1, (15, 3) for the minus strand of Experiment 1, (12, 4) for the minus strand of Experiment 2, and (11, 4) for the minus strand of Experiment 2, respectively. Any segment of the genome that passed these thresholds was termed the mined G4 (MG4) sequence. Within both strands of the human genome, we identified 1,054,941 MG4s and 936,545 MG4s in the two parallel experiments (Supplementary Table 6). Approximately 52% and 49% of all 356,298 predicted canonical quadruplexes were also detected by G4-miner and present in MG4s (Supplementary Table 6). Moreover, 736,689 of the total number of MG4s (70% for Experiment 1 and 78% for Experiment 2) were detected in both experiments (Fig. 3a). The high overlap between parallel experiments indicated that G4-miner is stable in quadruplex assigning.

**Structural analysis and genome browser of MG4 sequences**. We noticed that only 17% of the MG4s were predicted to be classically described G4s. Recently, noncanonical G4 structures such as long loops, bulges, and two quartets have been identified by structural biological and biophysical studies[31–33]. Therefore, we hierarchically assigned MG4 to five categories to elucidate the structural features (Fig. 3b). Canonical G4s, long loops, bulges, and two quartets accounted for 17%, 6%, 23%, and 34% of total MG4s based on the threshold of sequencing quality. Canonical G4 sequences with short loops were more easily identified than sequences with longer loops (Fig. 3c), which is consistent with their relative thermodynamic stability[32,33]. The MG4s in the "other" category does not contain known G4 structures, but also cause a slight and inconstant drop in sequencing quality. Sequence analysis revealed that most of them contained other DNA secondary structures, such as hairpin structures, poly-adenine/thymine, and i-motifs (Supplementary Table 7). To ensure fidelity, this category was removed for the following analyses. Our results revealed that G4-miner is able to mine and identify G4 sequences in the whole genome from ordinary sequencing data.

We quantified MG4s in exons, introns, promoters, untranslated regions (UTRs), and splicing junctions (Table 1). MG4s

were found to be densest in 5′-UTRs and splicing junctions, where G4s functioned in posttranscriptional regulation[34,35]. The density of G4s in genomic regions is consistent with the result obtained by adding K+ or PDS to sequencing buffers[24]. Visual inspection of genes rich in PG4s revealed that G4-miner is effective in mining G4 regions (Fig. 3d).

**Comparison with previously published data**. Compared to the result of the optimized G4-seq, which combined polymerase stop assays with next-generation sequencing[26], over 89% and 99% of PG4s detected by G4-miner were also detected by adding K+ and PDS to sequencing buffers, respectively (Fig. 4a). Meanwhile, 73% of PG4s detected by adding K+ to sequencing buffers were detected by G4-miner. The detected PG4s highly overlapped between adding K+ to the sequencing buffer and utilizing G4-miner. More PG4s are detected under G4-stabilizing conditions (PDS) than under standard sequencing conditions, which confirms the existing knowledge that G4 structures are thermodynamically stable in the presence of PDS. Over 40% of all MG4s were detected by adding PDS to sequencing buffers (Fig. 4b). In addition, 44% of the G4 sequences observed by adding K+ to sequencing buffers were detected in our method. We divided the G4 sequences into three parts and utilized structural analysis for each part (observed by both methods, only observed by G4-miner, and only observed under K+ or PDS conditions) (Fig. 4c, d). Unexpectedly, the most abundant structural category in the overlapping parts was canonical G4s, especially in the overlap under standard sequencing conditions and under K+ conditions (60%). Among noncanonical G4 structures, the bulge is the dominant category in the G4 sequences, which is only observed under the PDS condition (50%) and is the most abundant category under the K+ condition (36%). In fact, the bulge was the most abundant category in all G4 sequences detected under PDS conditions (45%) and was the second most abundant category in all G4 sequences under K+ conditions (30%; Supplementary Fig. 4). In contrast, two quartets were dominant in the G4 sequences only observed by G4-miner (52% for K+ and 58% for PDS), which was also the most abundant category in all G4 sequences detected by G4-miner (40%, Fig. 3b). The results show that both optimized G4-seq and G4-miner are reliable in detecting canonical G4s. However, the prevalence of detected G4 sequences by category varies between methods and detection conditions. Bulges are more likely to be detected under PDS conditions using G4-seq, and two quartets are prone to be seized using G4-miner.

Notably, the distribution of MG4 categories was highly consistent with the result of rG4-seq[25], which exploited reverse transcriptase stalling reactions containing K+ with G4-stabilizing PDS prior to sequencing (Supplementary Fig. 5). Two quartets are most abundant in the result of rG4-seq (K+–PDS), ~39% of detected G4 sequences, which is 40% in our method.

**Applicability evaluation of G4-miner by public datasets**. To evaluate the applicability of G4-miner, we tested the method by using public resequencing datasets of six species whose reference genomes are known. The sequencing depth of the selected species was at least ×50 (Supplementary Table 8). The density of PG4s varies among species, from 1.56 PG4s per million bases to 89.42 PG4s per million bases (Table 2). The threshold guanine content calculated for each genome ranged from 24 to 29% based on the criterion that 99% of PG4-positive segments were greater than the threshold. As a result, the detection rates of predicted canonical quadruplexes range from 32 to 58%. The results of *Homo sapiens* using public datasets are highly consistent with the results of local datasets, not only the number of detected PG4 but also the

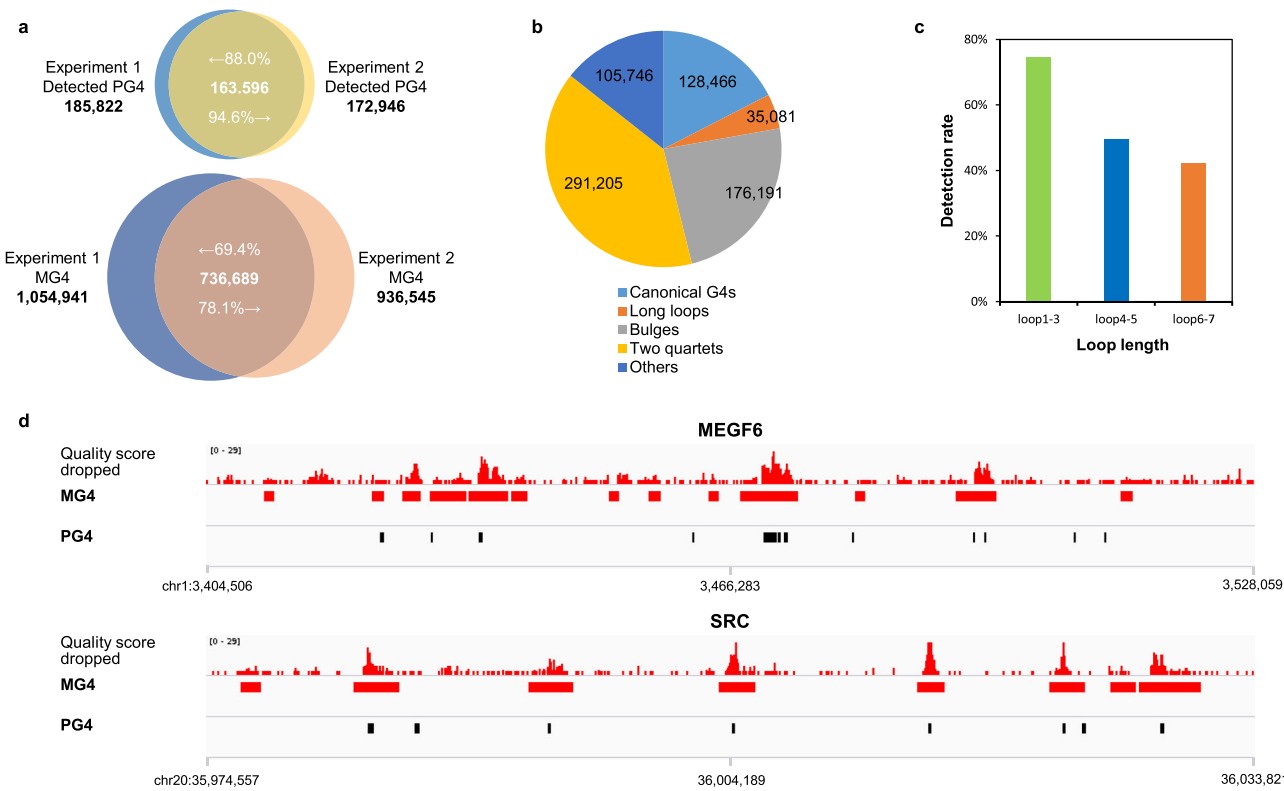

**Fig. 3 Analysis of mined PG4s. a** Overlap between PG4s that were detected in MG4s in two parallel runs and overlap between MG4s in two parallel runs. **b** Prevalence of MG4s by different G4 structural families. **c** Detection rate of canonical G4 sequences categorized by loop length. **d** Genome browser view of MG4 and PG4 across two oncogenes, MEGF6 and SRC. The median sequencing quality score dropped in each locus was tracked along the whole gene sequence. The regions above the threshold are shown as red bars (MG4s). The PG4s are shown as black bars.

**Table 1 Distribution of MG4s in different genomic regions.**

| Region | # Regions | Total region size[a] | # MG4s | MG4s density[b] | # Mined PG4s | Mined PG4s density[b] | # Canonical MG4s | Canonical MG4s density[b] |
|---|---|---|---|---|---|---|---|---|
| 3'-UTR | 46,297 | 38.3 | 6171 | 0.16 | 1565 | 0.04 | 2308 | 0.06 |
| 5'-UTR | 66,404 | 15.6 | 10,361 | 0.66 | 1776 | 0.11 | 5143 | 0.33 |
| Exon | 243,071 | 81.6 | 25,349 | 0.31 | 3664 | 0.04 | 11,513 | 0.14 |
| Intron | 211,307 | 1494.1 | 169,429 | 0.11 | 44,966 | 0.03 | 37,005 | 0.02 |
| CDS | 111,622 | 34.6 | 13,592 | 0.39 | 606 | 0.02 | 6397 | 0.18 |
| TSS 1000 up | 35,634 | 71.3 | 31,976 | 0.45 | 17,800 | 0.25 | 13,005 | 0.18 |
| TSS 1000 up down | 35,634 | 35.6 | 15,994 | 0.45 | 7670 | 0.22 | 6544 | 0.18 |
| Splice 50 | 385,429 | 38.5 | 19,748 | 0.51 | 4607 | 0.12 | 10,181 | 0.26 |

*CDS* coding region, *TSS 1000 up* 1000-base upstream of the transcription starting site, *TSS 1000 up down* 1000-base upstream and downstream of the transcription starting site.
[a]The total length of the specific kind of region (million base pairs).
[b]The density of the specific sequences in that region (# per kilobase pair).

distribution of MG4 categories. All kinds of G4 sequences were detected in the datasets of each species, and the distribution of MG4 categories was diverse among species (Supplementary Table 8). All the results demonstrated that G4-miner is reliable and widely applicable in quadruplex detection.

**Evaluation of the influence of SNVs on G4 formation.** SNVs are the differences at specific nucleotide locations in the genome and are one of the most important genetic variations among individuals. The alteration of single nucleotides might have different consequences at the phenotypic level[36]. Some of them might have different consequences in G4s, including controlling the formation, altering the structure, and influencing the stability. Therefore,

we inspected SNVs in the whole human genome of GM12878, both homozygous SNVs and heterozygous SNVs. For homozygous SNVs, we PG4s again using the modified genome, which altered nucleotides in specific locations. Due to the alternation of homozygous SNVs, 1502 new PG4s were predicted in this time, where 48% of them (728 PG4s) were identified as MG4s and 1077 existing PG4s were not predicted, where 60% of them (643 PG4s) were not recognized as MG4s (Experiment 1, Fig. 5a, b).

For heterozygous SNVs, we classified the sequencing reads into two groups based on allele types to identify MG4s separately. Among all heterozygous SNVs, 7.17% (186,904) showed different influences on the formation of G4s in the plus strand and 6.28% (163,666) in the minus strand (Fig. 5c and Table 3). All homozygous and heterozygous SNVs associated with G4s are

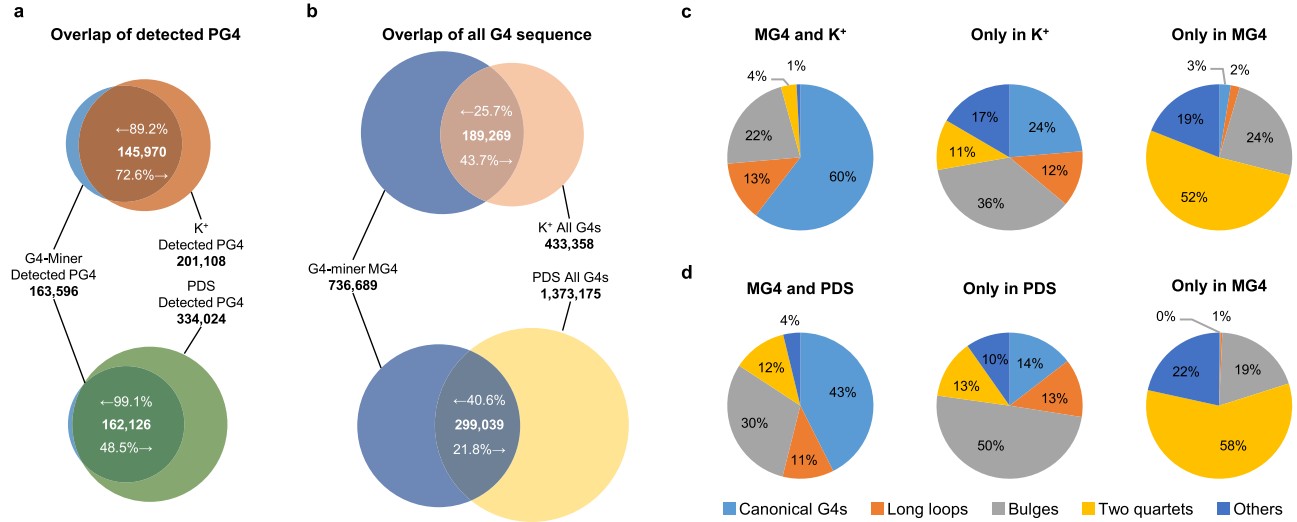

**Fig. 4 Comparison with the result of the optimized G4-seq. a** Overlap between PG4s that were detected in MG4s in standard sequencing buffers and K⁺ or PDS conditions[26]. **b** Overlap between MG4s in standard sequencing buffers and K⁺ or PDS conditions[26]. **c** Prevalence of G4 sequences detected both under K⁺ conditions and by G4-miner, only detected under K⁺ conditions and only detected by G4-miner. **d** Prevalence of G4 sequences detected both under PDS conditions and by G4-miner, only detected under PDS conditions and only detected by G4-miner.

**Table 2 MG4s detected for the six species using public datasets.**

| Species | Genome size (Mb) | # PG4s | Density of PG4s[a] (# PG4/Mb) | # MG4s | # PG4 detected in MG4 | % PG4 detected in MG4 | G-ratio |
|---|---|---|---|---|---|---|---|
| *Homo sapiens* | 3,095.7 | 356,298 | 57.55 | 790,108 | 145,420 | 40.81 | 0.28 |
| *Mus musculus* | 2,730.9 | 488,391 | 89.42 | 1,172,827 | 281,658 | 57.67 | 0.28 |
| *Drosophila melanogaster* | 143.7 | 10,036 | 34.92 | 57,760 | 4272 | 42.57 | 0.27 |
| *Arabidopsis Thaliana* | 119.7 | 1203 | 5.03 | 39,599 | 445 | 36.99 | 0.24 |
| *Caenorhabditis elegans* | 100.3 | 2154 | 10.74 | 13,471 | 689 | 31.99 | 0.25 |
| *Saccharomyces cerevisiae* | 12.2 | 38 | 1.56 | 2118 | 17 | 44.74 | 0.29 |

[a]Density of PG4s = #PG4s/(genome size ×2).

individual-specific G4s that might contain individual differences in chromatin architecture and many fundamental biological processes, such as DNA replication and gene regulation. Our results show that G4-miner can characterize G4s for specific individuals.

## Discussion

Many techniques have been developed to detect and map G4s, including chemical biology and genomic technologies. However, to date, most of these techniques' protocols are complicated by relying on G4-stabilizing conditions or G4 antibodies. To address this conundrum, we have established a convenient, high-throughput approach for identifying DNA G4 structures from ordinary next-generation sequencing data directly. Our results confirm that G4 structures are unstably formed in the standard sequencing buffer, which causes sequencing quality fluctuations. We conveniently identified canonical G4s and numerous noncanonical structures in the whole genome by inspecting quality fluctuations.

First, to discover the stability of G4-miner, we carried out two parallel experiments, and 736,689 MG4s were detected in both experiments. The high overlap between them indicated that G4-miner is stable in quadruplex assigning. Second, to verify the validity of the method, we compared the results of the G4-miner and the optimized G4-seq. The results of detected PG4s were highly consistent between these two approaches. Over 89 and 99% of PG4s detected by G4-miner were detected in the result of

optimized G4-seq, and 73% of PG4s detected under K⁺ conditions were seized by G4-miner. The comparison of all detected G4 sequences showed that optimized G4-seq and G4-miner are reliable in canonical G4 detection, but have a preference in detecting noncanonical G4s. G4-seq was preferred in detecting bulges, while G4-miner was preferred in detecting two quartets. The different preferences of noncanonical G4s of the methods can be used for different research purposes. Third, to evaluate the applicability of G4-miner, we tested the method by using public resequencing datasets of six species. The detection rates of the predicted canonical quadruplexes for different species are between 32 and 58%. The number of detected PG4 and the distribution of MG4 categories of *Homo sapiens* utilizing the public data is extremely similar to the results of the local data. The foregoing results demonstrate that the G-miner is adaptable in a wide range of situations.

G4-miner is a user-friendly approach because it uses standard sequencing data, which do not require the accommodation of a sequencing buffer. Although 45× sequencing depth for each strand was used, the PG4 detection rate plateaued when the sequencing depth was >20× for one strand (Supplementary Fig. 3). As at least two sequencing conditions are required for G4-seq, the sequencing cost of the G4-miner is lower. Considering the nonstandard sequencing reagents used by G4-seq, the competition in cost of G4-miner is more obvious. In addition, the principle of G4-miner can also be used to identify other structures

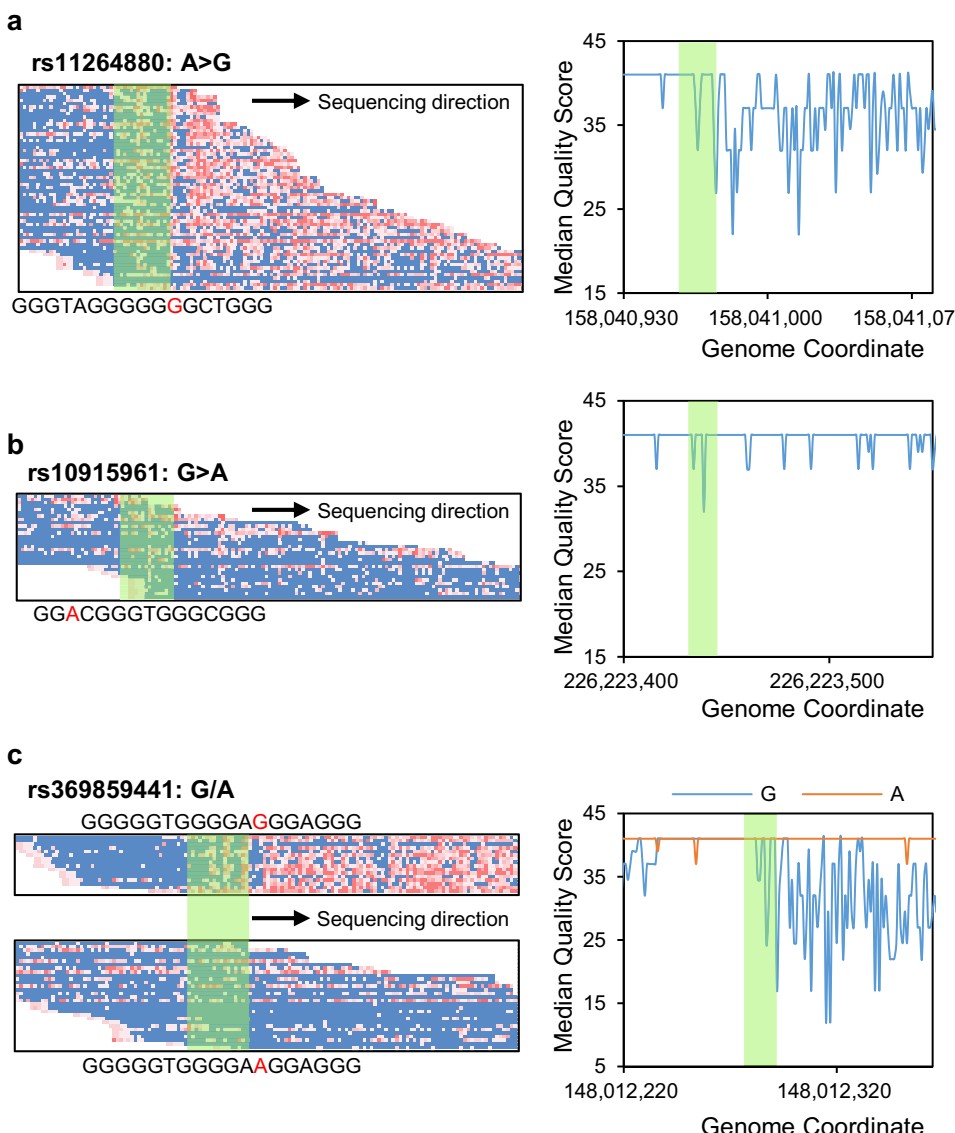

**Fig. 5 Influence of SNVs in G4 formation. a** The alteration of a homozygous SNV (rs11264880) causes the formation of a G4 structure and was detected. **b** The alteration of a homozygous SNV (rs10915961) prevents the formation of the G4 structure and was not detected. **c** Two alleles of the heterozygous SNV (rs369859441, G/A) have different influences on the formation of the G4 structure and were demonstrated by G4-miner (formed: allele G; not formed: allele A). The green shadows present the regions of the sequences.

### Table 3 The influence of heterozygous SNVs on MG4 detection.

| Strand | # Total heterozygous SNVs | # SNV (++)[a] | # SNV (+−) and SNV (−+)[b] | # SNV (−)[c] | # MG4 (+−) and MG4 (−+)[d] |
|---|---|---|---|---|---|
| Plus | 2,608,087 | 123,529 | 186,904 | 2,297,654 | 153,533 |
| Minus | | 109,410 | 163,666 | 2,335,191 | 134,532 |

[a]The number of heterozygous SNVs in which MG4s were identified in both groups of alleles.
[b]The number of heterozygous SNVs in which MG4s were identified only in the group with allele 1 or the group with allele 2.
[c]Number of heterozygous SNVs in which MG4s were not identified in either group of alleles.
[d]The number of heterozygous MG4s that were identified only in the group of allele 1 or group of allele 2.

that would influence sequencing quality and describe genomic maps of other secondary structures by their sequence characterization through quality fluctuations.

One key role of our method is to reveal the effect of SNVs on G4 formation. After inspecting SNVs in the whole human genome, we observed that the presence of some SNVs determines the formation of the G4 structure. The homozygous SNV causes the formation of the G4 structure or prevents the formation of the G4 structure. For

some heterozygous SNVs, the two alleles have different influences on the formation of G4 structure, as allele G caused quality fluctuations, and allele A did not. SNVs show significant individual differences, and the specific relation between SNVs and G4s might contain individual differences in chromatin architecture and fundamental biological processes. The G4-miner can be used to detect individualized G4 structures, which will accelerate the discovery of the relationship among SNVs, G4s, and biological processes.

In summary, our work provides G4-miner, a user-friendly and convenient method to identify G4 structures genome wide. The maps of G4s in the whole genome were obtained under a standard sequencing protocol by inspecting sequencing quality in the whole genome, seizing unexpected fluctuations locally, and ascribing some of the quality fluctuations to the unstable formation of G4. All the results demonstrate that G4-miner is sufficiently accurate or widely applicable in G4 sequence detection. In addition, by using G4-miner, we confirmed that some of the SNVs can influence the formation of G4 structures. The given approach is available to identify and characterize G4s genome-wide for specific individuals. This approach could also be extended to profile other DNA secondary structures that exist in a standard sequencing buffer.

## Methods

**Library construction and DNA sequencing for GM12878 cells**. Genomic DNA of GM12878 cells (Coriell Institute) was harvested using the QIAamp DNA Mini Kit (Qiagen). For each of the two parallel sequencing libraries, 5 μg genomic DNA was used to construct ~500–700 bp libraries according to the manufacturer's protocol (Illumina). The libraries were then subjected to paired-end sequencing on an Illumina HiSeq4000 platform (Illumina) with the 150 bp read option, according to the manufacturer's instructions.

**PG4 identification and region classification**. PG4s were predicted from hg19 using g4predict (https://github.com/mparker2/g4predict.git) according to the Quadparser algorithm[11] by searching for the "(G{3,}[ATCG]{1,7}){3}G{3,}" pattern. For the parameter determining step, the start site of the "positive region" was defined as the sequence up to 12 bases (approximate footprint of DNA polymerase) upstream of the PG4 start site, and the end site was 75 bases downstream of the start site. After removing the positive regions and 300 bases upstream and downstream of them, the remaining region was split into 75-base windows, which were defined as "negative regions."

**Sequencing alignment and processing**. Human genome sequence build 37 (hg19) downloaded from the UCSC genome browser was used as the reference for mapping. All qualified sequencing reads were mapped to hg19 using BWA-MEM (version 0.7.12-r1039) in paired-end mode[37]. The resulting SAM files were converted into BAM format, sorted based on genome position, and indexed using SAMtools (version 1.8)[38]. SNV calling was performed using the HaplotypeCaller with Genome Analysis Toolkit (GATK, version 4.0.3.0)[39]. Sequencing reads were analyzed using in-house Perl scripts.

**Low-quality region scanner**. For each sequencing read, the aligned sequence information was extracted from the BAM file. Reads were separated into forward strands and reverse strands according to the FLAG value. After the quality score was translated into decimal numbers, the median quality of every locus of different strands was calculated. The low-quality region scanner algorithm checked the median quality value of every locus sequentially. The parameter "M" is designed to describe the degree of low quality, and parameter "N" limits the number of low-quality bases to set the lower bound of the low-quality region. If the difference of median quality scores between one base and the just base upstream of it (the orientation of "up" or "down" is consistent with the orientation of the strand) is no less than M, the base with a higher quality value is "the flag site" whose quality score is defined as "the flag value," and the other is "low-quality site," also "the start site of the low-quality region." From the start site on, if the number of low-quality sites was greater than or equal to N in a range of 75 bp, a 75 nt low-quality region was marked as a "hit."

**Parameter determining and MG4 detection**. Positive regions (with PG4s) and negative regions (without PG4s) from the "PG4 identification and region classification" step were used to calculate the hit ratio according to the previous low-quality region scanning algorithm. The parameters M and N were set as 1–12 and 1–20, respectively. Under each M–N set, a positive rate and a false-positive rate were calculated. For both strands of two replicates, parameter sets that can obtain the maximum positive rate while keeping the false-positive rate lower than 1% were used in the following steps. To eliminate the influence of other non-G4 secondary structures, the ratio of G/C bases in detection regions should be at least 28% (99% of positive regions contain 28% or more G/C bases). For the whole genomic analysis, low-quality regions from two strands of two replicates were scanned independently under their most suitable parameter sets. For the more complete detection of positive regions, the low-quality regions in the MG4 detection step extended 35 bases (the longest length of canonical G4s) upstream of the start site. Low-quality regions with overlapping fragments were merged into a single region. The merged regions are considered MG4s.

**Structure analysis of MG4 categories**. MG4s were stratified into different categories according to the patterns of G4s. To avoid calculating the same MG4s multiple times, we followed the order based on the predicted stability of different patterns: canonical > long loops > bulges > two tracts > others. These different categories are defined as follows: canonical, (G{3,}N{1,7}){3}G{3,}, with N = [ATCG]; long loops, G{3,}N{8,12}(G{3}N{1,7}){2}G{3,} or G{3,}N{1,7}G{3,}N{8,12}G{3,}N{1,7}G{3,} or (G{3,}N{8,12}){2}G{3}N{1,7}G{3,} or G{3,}N{8,12}G{3,}N{1,7}G{3,}N{8,12}G{3,} or (G{3,}N{8,12}){3}G{3,}; bulges, G4 with any G-tract being GH{1,7}GG or multi G-tracts being GH{1,2}GG, with H = [ATC]; two tracts, (G{2}N{1,12}){3}G{2}; and others that were not in a previous category. MG4s with more than one pattern would only be categorized as more stable MG4s. The "others" category was further stratified into subcategories containing non-G4 structures (e.g., i-motifs, hairpins, repeats, poly-A/T). Some of these categories are defined as follows: i-motif, (C{3,}N{1,7}){3}C{3,}; hairpins, stem-loop structures with stems no <7 bp and loops no >30 bp; poly-A/T, continuous 10 or more A/T bases.

**Comparison of previously published data**. To compare the results with those of the optimized G4-seq[26], whole-genome mapping results of DNA G4s in Homo sapiens by G4-seq were downloaded from the NCBI Gene Expression Omnibus (GEO; http://www.ncbi.nlm.nih.gov/geo/; accession numbers GSM3003539 and GSM3003540). G4s were stratified into different categories using the pipeline described above.

**Genomic regions analysis**. Gene annotation files were downloaded from the UCSC genome browser website (https://genome.ucsc.edu/), genome version hg19. Genomic regions contain 5′-UTRs, 3′-UTRs, exons, introns, CDSs, promoter regions, TSS (translational start site) regions, and splice regions. TSS regions are defined as the 1000 bp upstream and downstream of the TSS. Promoters are 1000 bases upstream TSS. Splice regions are 50 base upstream and downstream splice sites. For these regions, their total number and size were calculated. The number of MG4s, including total MG4s and canonical MG4s, and predicted PG4s overlapping with the region intervals were calculated. The number of MG4s was the intersection of two replications. For each gene annotated in the human genome (hg19), the number of predicted PG4s of MG4s (PG4inMG4s) in two replications was counted. The density of G4s was calculated by dividing the respective counts by the length of genes or regions and multiplying by 1000.

**The applicability analysis of the G4-miner using public data**. The applicability of the G4-miner was further evaluated by using data from public databases. The sequencing data of the six species (Homo sapiens, Mus musculus, Drosophila melanogaster, Arabidopsis thaliana, Caenorhabditis elegans, and Saccharomyces cerevisiae) whose reference genomes are known were retrieved from the Sequence Read Archive (SRA) in FASTQ format. The reference genomes of the six species were downloaded from the UCSC Genome Browser website (https://genome.ucsc.edu/) or NCBI Genome resources (https://www.ncbi.nlm.nih.gov/home/genomes/) (Supplementary Table 9).

The bioinformatic analysis pipeline for the public data is the same as that of in-house data. Specifically, the filtering thresholds of guanine content were calculated for each genome based on the criterion that 99% of PG4-positive segments were greater than the threshold. The ratio of G/C bases in detection regions should be higher than the thresholds to eliminate the influence of other non-G4 secondary structures.

**Reporting summary**. Further information on research design is available in the Nature Research Reporting Summary linked to this article.

## Data availability

The processed data reported in this paper have been deposited in the NCBI Gene Expression Omnibus (GEO) under accession number GSE159307. Underlying sequencing data are available at the NCBI Sequence Read Archive under accession number SRP286586. The reference genomes of the six species are available on the UCSC Genome Browser website (Homo sapiens, Mus musculus, Drosophila melanogaster, and Saccharomyces cerevisiae, https://genome.ucsc.edu/) or NCBI's Genome resources (Arabidopsis Thaliana, Caenorhabditis elegans, https://www.ncbi.nlm.nih.gov/home/genomes/) (Supplementary Table 9).

## Code availability

The bioinformatics pipeline and all custom codes have been deposited in the Github repository (https://github.com/tulabcode/G4-miner, https://doi.org/10.5281/zenodo.5516275)[40].

# ARTICLE

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

## Acknowledgements

This work was supported by the Natural Science Foundation of Jiangsu Province (BK20211513), National Natural Science Foundation of China (61972084, 61971125), and Six Talent Peaks Project of Jiangsu Province (2019-SWYY-004).

## Author contributions

M.D., W.L. and Y.Z. implemented and performed the analysis. J.T., X.S. and Z.L. designed the experiments and analysis. J.T., M.D. and N.L. interpreted the results and co-wrote the manuscript. J.T. and Z.L. reviewed and revised the manuscript.

## Competing interests

The authors declare no competing interests.
