## [Peer Review File · Nature Communications]

Direct genome-wide identification of G-quadruplex structures by whole-genome resequencingReviewers' Comments:

Reviewer #1:

Remarks to the Author:

In this manuscript Jing Tu et al. describe a method to detect DNA G-quadruplex secondary structures within the human genome using standard Illumina sequencing data, exploiting fluctuation in sequencing quality score that are triggered by G4 structures present in the sequencing reads.

There have been many computational and experimental methods to map G4s within the human genome to date, pointing towards a significant issue in terms of novelty, as most of the observations reported in this manuscript basically a confirmation of what observed with previous methods with little, if any, new insights on G4-prevalence in the human genome revealed. Furthermore, this method is highly reminiscent of G4-seq, a sequencing-based method to detect G4 across different genomes, with the difference that G4-seq relies on changing sequencing conditions in order to promote or prevent G4-formation and enrich for G4-selective events, whilst this method simply relies on drop of quality score that can be triggered by non G4-mediated effects, making this map far less reliable than the one currently available.

Besides the novelty concerns that are perhaps the most obvious, there are also some technical concerns related on how the authors analyse their dataset and validate their method (I will elaborate more on this below) and the English quality is very poor, with grammar mistakes present even in the title of the manuscript. Overall, I cannot recommend publication of this manuscript in Nat Commun, mainly due to the significant lack of novelty, poor scholarly presentation and major technical issues that I articulate below:

1) The authors are correct when they state that the G4-seq method cannot be easily reproduced by the community owing to the lack of accessibility of buffer, that are directly provided by Illumina. Their method on the other hand can overcome this limitation as it simply looks at reads sequenced under standard Illumina sequencing conditions, but on the other hand the drop in sequencing quality might be easily due to non G4-mediated effects, as pointed out by the poor overlap between MG4 (this method) and OG4 (G4-seq). Moreover, if the main strength of the method is to allow G4-mapping to users that cannot apply G4-seq, why applying it to the human genome that has been already mapped by G4-seq? The authors should exploit it to provide novel G4-maps that are not currently available to the community, otherwise what is the point of using a less selective and sensitive method to map a genome that has been already characterised by a better method? If the authors want to really showcase the utility of their approach they should cover a genome that has not been characterised by G4-seq

2) In table 6 is already clear that this method is not as sensitive and selective as G4-Seq to detect G4s, as the authors obtain a 52% and a 49% of overlap of MG4 peaks with predicted G4 sequences (PG4). This is in stark contrast to the 73% obtained in the G4-seq seminal paper and the authors fail to comment of this. This possibly reflects the detection on non-G4 mediated quality drop during their method which fails to enrich specifically G4s as the conditions change in G4-seq instead ensures

3) When the authors comment of the results and how they compare to G4-seq in line 189 they refer to 75% and 82% overlap between the two methods. However, this is highly misleading as it is limited to PG4 sequences, which represent a very small proportion of their entire dataset and, therefore, bias this analysis significantly. In G4-seq most of the non-canonical G4s required highly stabilising conditions to be detected and is highly unlikely that those are detected by this method. The real overlap between the methods is modest and is worth nothing that there are more than twice of the amount of "other" G4s detected with this method compared to G4-seq, which is concerning and indicative of non-G4 effects driving the detection of peaks.

4) Lastly the quality of the English is not appropriate for publication, especially in a Nature group

journal and this should be thoroughly revised prior resubmission anywhere else.

Reviewer #2:

Remarks to the Author:

The manuscript by Tu et al. proposes a simple method to identify G-quadruplex (G4) structures from whole genome sequencing datasets. It starts with the observations that sequencing quality drops at loci forming secondary DNA G-quadruplexes structures that were previously validated. From there, they have developed the G4-miner tool, which gives comparable outputs to the more complicated technology G4-seq. It appears also efficient in detecting allele specific G4 structures at polymorphic sites.

While apparently simple in its concept and implementation, this technology could fill a gap in our comprehension in G-quadruplex biochemistry and biology. To date, technologies mapping G4 require the use of alternative buffers which are almost unfeasible for most laboratories (G4-seq), or the need of G4 binding nanobodies for in vivo or in cellulo context studies (G4-ChIP-seq). These techniques have been so far successfully performed by virtually only one laboratory. The community would really benefit from transferable technology for genome-wide G4 detection. In general, the work presented is of significant relevance to the field. However, the reproducibility of the method from other sequencing facilities is difficult to evaluate due to the limited amount of presented dataset. In addition, the manuscript must be largely improved in its presentation and description. In conclusion, this study should be further developed before a publication in Nature communications.

Major comments:

1. There are only two replicates of whole genome sequencing (WGS) samples used in the study. To allow drawing conclusions from the study and supporting the robustness of the method, the G4-miner tool should be tested on other publically available samples. There are many WGS datasets that are publically available (eg. Encode consortium).

2. Authors must clearly state which dataset were used in the study and how they were processed and produced.

- Figures 1 and 2 show whole genome sequencing datasets. I suppose they have been produced in this study. If yes, this should be mentioned in the text before.

- Figure 3c-d show "overlap between MG4s in standard sequencing buffers and K⁺ or Na⁺ + PDS conditions". MG4s were only mapped in standard sequencing buffers. Thus, I deduce that means overlap between MG4s and G4-seq detected G-quadruplexes from previously published work. This should be clarified. Were the previously published dataset re-analysed?

- Sequencing conditions, alignment software and data processing prior G4-miner must be indicated in the Methods section.

3. In general, figure and table legends need to be more detailed.

- Figure 2, the legend should mention that a) and b) depict regions of validated G4 forming sequences and that c) and d) are negative controls.

- Figure 3 legends must clearly indicate which datasets were used.

- Figure 4c, could the authors please indicate coordinates of the genomic location depicted and add gene annotations on the figure (gene start, end, exon, intron...)

- Table 1, please detail acronyms used in the table. CDS as coding sequence, what is TSS 1000 up down, how does it compare to Promoter 1000 up (which is probably the region of 1000 bp upstream TSSs). Those should be defined.

- Table 2, the total of heterozygous SNVs is mentioned in column 2. From those either 2 MG4s (one on each alleles), 1 MG4 (only on one allele) or none were found (column 3-5). What is the last column? How do these numbers compare to the others? As stated and described, this column cannot be

understood.

- All supplementary tables and figures need legends.

Minor comments:

1. Line 58-59, authors comment that G4-seq needs to be sequenced twice and such is twice as much expensive than technologies that would only need 1 run of sequencing. However, in its first set up (Chambers et al. 2015), G4-seq were sequenced in duplicates in 3 different conditions at ~300 millions reads (6x300 ~1.8 billions reads). 1.8 billions reads (150 bp paired-end) is exactly what has been used in this study to support the technology. Authors should comment on this in the discussion section since their technology remains very expensive.

2. Figures 1, 4 and 5 show that the decrease in sequencing quality varies between G-quadruplexes that are mapped. Authors could strengthen the G4-miner tool by testing if these differences arise from differences in the G-quadruplexes stability. This could be achieved by testing quality scores vs model G-quadruplexes with defined in vitro stability.

3. Supplementary tables 2 and 3 would gain as being presented as charts.

4. Lines 153-157, authors mention that the technology supports that the human genome contains more G-quadruplexes than predicted-G4s. Could the authors specifically analyse the composition of the non-predicted G4s?

5. Supplementary table 4: could the authors discuss why quality score dropped and loci reached thresholds criteria vary between the samples and the strands of the experiments.

6. Line 222, the acronym OQs should be defined.

7. Line 137-140, the algorithm used for PG4 definition should be mentioned (Quadparser).

Reviewer: 1

- 1) The authors are correct when they state that the G4-seq method cannot be easily reproduced by the community owing to the lack of accessibility of buffer, that are directly provided by Illumina. Their method on the other hand can overcome this limitation as it simply looks at reads sequenced under standard Illumina sequencing conditions, but on the other hand the drop in sequencing quality might be easily due to non G4-mediated effects, as pointed out by the poor overlap between MG4 (this method) and OG4 (G4-seq). Moreover, if the main strength of the method is to allow G4-mapping to users that cannot apply G4-seq, why applying it to the human genome that has been already mapped by G4-seq? The authors should exploit it to provide novel G4-maps that are not currently available to the community, otherwise what is the point of using a less selective and sensitive method to map a genome that has been already characterised by a better method? If the authors want to really showcase the utility of their approach they should cover a genome that has not been characterised by G4-seq

[Reply]: Thanks for your great suggestion on improving the accessibility of our manuscript. The manuscript has been revised accordingly to clarify the above concerns. To evaluate the applicability of G4-miner, we tested the method by using the public resequencing datasets of 6 species whose reference genomes are known. Based on the results of the detection process developed in this research, the detection rate of the predicted canonical quadruplexes for different species is between 32% and 58%. Furthermore, not only the number of detected PG4, but also the distribution of MG4 categories, the results of Homo sapiens utilizing the public data set are extremely similar with the results of the local data set. The foregoing results demonstrate that the G-miner developed in this study is adaptable in a wide range of situations. The relevant research results have been detailed in the "Applicability evaluation of G4-miner by public datasets" chapter of the revised manuscript, with screenshots for viewing:

Applicability evaluation of G4-miner by public datasets

To evaluate the applicability of G4-miner, we tested the method by using the public resequencing datasets of 9 species whose reference genomes are known. The sequencing depth of the selected species is at least $50\times$ (**Supplementary Table 8**). The density of PG4s varies among species, from 1.56 PG4s per million bases to 89.42 PG4s per million bases (**Table 2**). The threshold guanine content calculated for each genome range from 24% to 29%, based on the criterion that 99% PG4 positive segments are greater than the threshold. As a result, the detection rate of predicted canonical quadruplexes is from 32% to 58%. The results of homo sapiens using public datasets are highly consistent with the results of local datasets, not only the number of detected PG4, but also the distribution of MG4 categories. All kinds of G4 sequences are detected in the datasets of each species, and the distribution of MG4 categories is diverse among species (**Supplementary Table 8**). All the results demonstrated that G4-miner is reliable and widely applicable in quadruplex detection.

Table 2 MG4s detected for the 6 species using the public datasets.

Species	Genome size (Mb)	# PG4s	Density of PG4s ¹ (# PG4/Mb)	# MG4s	# PG4 detected in MG4	% PG4 detected in MG4	G-ratio
Homo sapiens	3,095.7	356,298	57.55	790,108	145,420	40.81%	0.28
Mus musculus	2,730.9	488,391	89.42	1,172,827	281,658	57.67%	0.28
Drosophila melanogaster	143.7	10,036	34.92	57,760	4,272	42.57%	0.27
Arabidopsis Thaliana	119.7	1,203	5.03	39,599	445	36.99%	0.24
Caenorhabditis elegans	100.3	2,154	10.74	13,471	689	31.99%	0.25
Saccharomyces cerevisiae	12.2	38	1.56	2,118	17	44.74%	0.29

¹ Density of PG4s = #PG4s / (Genome size × 2)

Furthermore, as we mentioned in the original manuscript, some of SNVs might have different consequences in G4s, including controlling the formation, altering the structure and influencing the stability. Therefore, although human genome has been already mapped, it is still meaningful to characterize these individual G4s.

- 2) In table 6 is already clear that this method is not as sensitive and selective as G4-Seq to detect G4s, as the authors obtain a 52% and a 49% of overlap of MG4 peaks with predicted G4 sequences (PG4). This is in stark contrast to the 73% obtained in the G4-seq seminal paper and the authors fail to comment of this. This possibly reflects the detection on non-G4 mediated quality drop during their method which fails to enrich specifically G4s as the conditions change in G4-seq instead ensures.

[Reply]: We appreciate your feedback and professional review work on our article. To improve our work, we compared the results of the optimized G4-seq which was newly published (*Nucleic Acids Research*, 2019, 47, 3862–3874.) with the results of this study instead of the results of the original G4-seq (*Nature Biotechnology*, 2015, 33, 877–881.). The results of the optimized G4-seq are proved to be more accurate than that of original G4-seq. Comparing results in the revised manuscript make our results more persuasive, as shown in **Figure 1**. It can be seen from the figure that compared with the optimized data, the overlap of the detected PG4 is higher. Over 89% and 99% of PG4s detected by G4-miner are also detected by adding K⁺ and PDS to sequencing buffers, separately. Meanwhile, 73% of PG4s detected by adding K⁺ to sequencing buffers are detected by G4-miner. G4-miner obtained 45.9% of all PG4s, which is close to the result of G4-seq under K⁺ condition (56.4%). The results under PDS condition are not appropriate to compare with the results under ion conditions, as PDS is a ligand which stabilized G4 sequences.

Figure 1 Comparison of the degree of overlap of detected PG4

- 3) When the authors comment of the results and how they compare to G4-seq in line 189 they refer to 75% and 82% overlap between the two methods. However, this is highly misleading as it is limited to PG4 sequences, which represent a very small proportion of their entire dataset and, therefore, bias this analysis significantly. In G4-seq most of the non-canonical G4s required highly stabilising conditions to be detected and is highly unlikely that those are detected by this method. The real overlap between the methods is modest and is worth nothing that there are more than twice of the amount of “other” G4s detected with this method compared to G4-seq, which is concerning and indicative of non-G4 effects driving the detection of peaks.

[Reply]: Thanks for your great suggestion on improving the accessibility of our manuscript. Firstly, the overlap between this method and the optimized G4-seq has increased to 89% and 99% as shown in **Figure 1**. Additionally, we compared all the G4 sequences detected by these methods. As a result, 44% of the G4 sequences observed by adding K^+ in sequencing buffers were detected in our method. We divided the G4 sequences into three parts and utilized structural analysis for each part as shown in the **Figure 2** (observed by both two methods, only observed by G4-miner, and only observed under K^+ or PDS condition). It can be seen from the figure that G4-seq and this study exhibit the highest degree of overlap in the detection results of canonical G4. However, the two methods have different preferences. G4-seq is more sensitive to bulges while this study is more sensitive to the two quartets. In addition, our classification results of the G4 structure are highly consistent with the results of rG4-seq (**Figure 3**, *Nature Methods*, 2016, 13, 841–844.). A more detailed description was added to the "**Comparison with previously published data**" section of the revised manuscript. The comparison results show that our method and G4-seq are different in preference and can be used for different research purposes. Therefore, this research still makes a valuable contribution to the field.

Figure 2 The prevalence of G4 sequences detected under different conditions

Figure 3 The prevalence of G4 sequences detected under different conditions

4) Lastly the quality of the English is not appropriate for publication, especially in a Nature group journal and this should be thoroughly revised prior resubmission anywhere else.

[Reply]: Thanks for your suggestion. We have polished our manuscript, corrected known typo and grammar errors. We marked them in the revised paper using revision mode. We appreciate for Editors/Reviewers' warm work earnestly and hope that the correction will meet with approval.

Reviewer: 2

Major comments:

- 1) There are only two replicates of whole genome sequencing (WGS) samples used in the study. To allow drawing conclusions from the study and supporting the robustness of the method, the G4-miner tool should be tested on other publically available samples. They are many WGS datasets that are publically available (eg. Encode consortium).

[Reply]: Thank you for your valuable suggestions to improve the quality of our manuscript. We have added the suggested content to the manuscript on "**Applicability evaluation of G4-miner by public datasets**". In this section, we tested the method by using the public resequencing datasets of 6 species whose reference genomes are known. All the results demonstrated that G4 miner is reliable and widely applicable in quadruplex detection, with screenshots for viewing:

Applicability evaluation of G4-miner by public datasets

To evaluate the applicability of G4-miner, we tested the method by using the public resequencing datasets of 9 species whose reference genomes are known. The sequencing depth of the selected species is at least $50\times$ (**Supplementary Table 8**). The density of PG4s varies among species, from 1.56 PG4s per million bases to 89.42 PG4s per million bases (**Table 2**). The threshold guanine content calculated for each genome range from 24% to 29%, based on the criterion that 99% PG4 positive segments are greater than the threshold. As a result, the detection rate of predicted canonical quadruplexes is from 32% to 58%. The results of homo sapiens using public datasets are highly consistent with the results of local datasets, not only the number of detected PG4, but also the distribution of MG4 categories. All kinds of G4 sequences are detected in the datasets of each species, and the distribution of MG4 categories is diverse among species (**Supplementary Table 8**). All the results demonstrated that G4-miner is reliable and widely applicable in quadruplex detection.

Table 2 MG4s detected for the 6 species using the public datasets.

Species	Genome size (Mb)	# PG4s	Density of PG4s ¹ (# PG4/Mb)	# MG4s	# PG4 detected in MG4	% PG4 detected in MG4	G-ratio
Homo sapiens	3,095.7	356,298	57.55	790,108	145,420	40.81%	0.28
Mus musculus	2,730.9	488,391	89.42	1,172,827	281,658	57.67%	0.28
Drosophila melanogaster	143.7	10,036	34.92	57,760	4,272	42.57%	0.27
Arabidopsis Thaliana	119.7	1,203	5.03	39,599	445	36.99%	0.24
Caenorhabditis elegans	100.3	2,154	10.74	13,471	689	31.99%	0.25
Saccharomyces cerevisiae	12.2	38	1.56	2,118	17	44.74%	0.29

¹ Density of PG4s = #PG4s / (Genome size \times 2)

- 2) Authors must clearly state which dataset were used in the study and how they were processed and produced.
 - Figures 1 and 2 show whole genome sequencing datasets. I suppose they have been produced in this study. If yes, this should be mentioned in the text before.
 - Figure 3c-d show “overlap between MG4s in standard sequencing buffers and K⁺ or Na⁺ + PDS conditions”. MG4s were only mapped in standard sequencing buffers. Thus, I deduce

that means overlap between MG4s and G4-seq detected G-quadruplexes from previously published work. This should be clarified. Were the previously published dataset re-analysed? - Sequencing conditions, alignment software and data processing prior G4-miner must be indicated in the Methods section.

[Reply]: We feel great thanks for your professional review work on our article. As you are concerned, there are several problems that need to be addressed. According to your nice suggestions, we have added data sources, sequencing conditions, calibration software and data processing methods in the method section, and screenshots are available for browsing:

Library construction and DNA sequencing for GM12878 cells

Genomic DNA of GM12878 cells (Coriell Institute) was harvested by the use of the QIAamp DNA Mini Kit (Qiagen). For each of the two parallel sequencing library, 5 µg genomic DNA was used to construct approximately 500-700 bp libraries according to the manufacturer's protocol (Illumina). The libraries were then subjected to paired-end sequencing on an Illumina HiSeq4000 platform (Illumina) with the 150 bp read option, according to manufacturer's instructions.

Sequencing alignment and processing

Human genome sequences build 37 (hg19) downloaded from UCSC genome browser was used as the reference for mapping. All qualified sequencing reads were mapped to hg19 using the BWA-MEM (version 0.7.12-r1039) in paired-end mode³⁷. The resulted SAM files were converted into BAM format, sorted based on genome position, and indexed using SAMtools (version 1.8)³⁸. The single nucleotide variant calling was performed using the HaplotypeCaller with Genome Analysis Toolkit (GATK, version 4.0.3.0)³⁹. Sequencing reads were analyzed using in-house Perl scripts.

Comparison of previously published data

To compare the results with that of the optimized G4-seq²⁶, whole genome mapping results of DNA G-quadruplexes in homo sapiens by G4-seq were download from NCBI Gene Expression Omnibus (GEO; <http://www.ncbi.nlm.nih.gov/geo/>; accession number GSM3003539, GSM3003540). G-quadruplexes were stratified into different categories using the pipeline described above.

The applicability analysis of the G4-miner using public data

The applicability of the G4-miner was further evaluated by using the data from public databases. The sequencing data of the 6 species (*Homo sapiens*, *Mus musculus*, *Drosophila melanogaster*, *Arabidopsis thaliana*, *Caenorhabditis elegans*, and *Saccharomyces cerevisiae*) whose reference genome are known were retrieved from the Sequence Read Archive (SRA) in FASTQ format. The reference genomes of the 6 species were downloaded from the UCSC Genome Browser website (<https://genome.ucsc.edu/>) or NCBI's Genome resources (<https://www.ncbi.nlm.nih.gov/home/genomes/>) (Supplementary Table 7).

The bioinformatic analysis pipeline for the public data is the same as that of in-house data. Specifically, the filtering thresholds of guanine content were calculated for each genome, based on the criterion that 99% PG4 positive segments are greater than the threshold. The ratio of G/C bases in detection regions should be higher than the thresholds to eliminate the influence of other non-G-quadruplex secondary structures.

To be noticed, instead of comparing the results of the original G4-seq (*Nature Biotechnology*, 2015, 33, 877–881.) with the results of this study, we compared the results of the optimized G4-seq (*Nucleic Acids Research*, 2019, 47, 3862–3874.) with the results of this research in the revised manuscript. The results of the optimized G4-seq are proved to be more accurate than that of original G4-seq.

- 3) In general, figure and table legends need to be more detailed.
- Figure 2, the legend should mention that a) and b) depict regions of validated G4 forming sequences and that c) and d) are negative controls.
 - Figure 3 legends must clearly indicate which datasets were used.
 - Figure 4c, could the authors please indicate coordinates of the genomic location depicted and add gene annotations on the figure (gene start, end, exon, intron...)
 - Table 1, please detail acronyms used in the table. CDS as coding sequence, what is TSS 1000 up down, how does it compare to Promoter 1000 up (which is probably the region of 1000 bp upstream TSSs). Those should be defined.
 - Table 2, the total of heterozygous SNVs is mentioned in column 2. From those either 2 MG4s (one on each alleles), 1 MG4 (only on one allele) or none were found (column 3-5). What is the last column? How do these numbers compare to the others? As stated and described, this column cannot be understood.
 - All supplementary tables and figures need legends.

[Reply]: We are very grateful to your comments and thoughtful suggestions. All figures and table legends have been adjusted to conform to the requirements. The relevant contents are provided below as a screen dump for your quick reference.

- Figure 2

Based on your suggestion, we have made a more detailed description of the legend as follows:

Figure 2 Sequencing quality of four known controls. (a) src, a sequence containing stable G4 structure; (b) myc, a sequence containing stable G4 structure; (c) a-repeat, negative control; (d) g-rich, negative control. The yellow shadows present the regions of the sequences. ↵

- Figure 3

Thank you for your reminding. We feel really sorry for our careless mistakes. We used the G4-seq data (Nature Biotechnology, 2015, 33, 877–881.) in the original manuscript, but we used the optimized G4-seq data (Nucleic Acids Research, 2019, 47, 3862–3874.) in the revised manuscript to make our results more convincing. The analysis of the relevant results has been explained in the previous question and elaborated in the chapter "**Comparison with previously published data**".

- Figure 4c

Thank you for your careful review. Based on your comments, we have made the corrections to make the figure clearer within the whole manuscript.

- Table 1

The abbreviations of all professional terms in the table have been attached as footnotes under the main body of the table, and screenshots are available for viewing.

¹ The total length of the specific kind of region (million base pairs). ↵
² The density of the specific sequences in that region (# per kilo base pairs). ↵
³ CDS, coding region. ↵
⁴ TSS 1000 up, 1000 bases upstream the transcription starting site. ↵
⁵ TSS 1000 up down, 1000 bases up- and down-stream the transcription starting site.

- Table 2

Thank you for your careful review. We apologize for not describing the legend clearer. In fact,

the fourth column is based on the results of heterogeneous SNV statistics, and the last column is based on the results of MG4 statistics. The reason for the difference in quantity is that some MG4s can cover 2 or more SNVs. Therefore, the number of MG4 is less than that of SNV.

Minor comments:

- 1) Line 58-59, authors comment that G4-seq needs to be sequenced twice and such is twice as much expensive than technologies that would only need 1 run of sequencing. However, in its first set up (Chambers et al. 2015), G4-seq were sequenced in duplicates in 3 different conditions at ~300 millions reads (6x300 ~1.8 billions reads). 1.8 billions reads (150 bp paired-end) is exactly what has been used in this study to support the technology. Authors should comment on this in the discussion section since their technology remains very expensive.

[Reply]: We thank you very much for your comments for pointing out this omission. You are right that the cost is comparable between G4-seq and our method. It is calculated only based on all sequencing data we generate. But as shown in **supplementary Figure 5** of our manuscript, PG4 detection rate got plateau when sequencing depth was greater than 20× for one stand. About 840 million reads are required when sequencing depth is 20× for one stand, about half of that G4-seq required. Besides, considering the non-standard sequencing reagents used by G4-seq, our method might be competitive in cost. More importantly, the user friendliness is the core advantage of G4-miner as G4-seq requires to accommodate sequencing buffer, and no commercial kits is available up till now. The pointed omission is very important, we revised the introduction and add comments on this in the discussion section according to your suggestion.

- 2) Figures 1, 4 and 5 show that the decrease in sequencing quality varies between G-quadruplexes that are mapped. Authors could strengthen the G4-miner tool by testing if these differences arise from differences in the G-quadruplexes stability. This could be achieved by testing quality scores vs model G-quadruplexes with defined in vitro stability.

[Reply]: Thank you for this suggestion. Maybe Figure 3c in the manuscript could partially answer this question. As shown in this figure, the detection rates of canonical G4s decreased along loop length, which support the known knowledge that the stability of canonical G4s is negative correlated with loop length. Also, testing quality scores vs model G-quadruplexes can define in vitro stability of G4s. But G4-miner maybe not precise enough to give answers in the present form. It is valuable to investigate the in vitro stability of G4s. We will try our best to improve the precision in stability definition of our method and try to give answers in the future. Thank you again for your positive comments and valuable suggestions to improve the quality of our manuscript.

- 3) Supplementary tables 2 and 3 would gain as being presented as charts.

[Reply]: Thank you again for your positive comments and valuable suggestions to improve the quality of our manuscript. The specific section has been updated according to your suggestions.

- 4) Lines 153-157, authors mention that the technology supports that the human genome contains more G-quadruplexes than predicted-G4s. Could the authors specifically analyse the composition of the non-predicted G4s?

[Reply]: According to your suggestions, we have supplemented relevant discussions in the revised manuscript, screenshots are available for review:

loops (Fig. 3c), which is keeping with their relative thermodynamic stability^{32,33}. The MG4s in 'other' category do not contain known G4 structures, but also cause slight and inconstant drop of sequencing quality. The sequences analysis revealed that most of them contained other DNA secondary structures, such as hairpin structure, poly-adenine/thymine and i-motif (Supplementary Table 7). To ensure fidelity, this category was removed for the following analyses. Our results revealed that the G4-miner is able to mine and identify G4 sequences in whole genome from ordinary sequencing data. »

5) Supplementary table 4: could the authors discuss why quality score dropped and loci reached thresholds criteria vary between the samples and the strands of the experiments.

[Reply]: We sincerely thank the reviewer for careful reading. The quality of sequencing will fluctuate throughout the experiment, mainly caused by secondary structures such as G4. There are slight fluctuations in each run due to the influence of the optical sensor system of the sequencer and the reliability of chemical reagents. And because the cutoff we choose is discrete rather than continuous, the impact of the above fluctuations may be amplified. We hope our explanation can solve your doubts.

6) Line 222, the acronym OQs should be defined.

[Reply]: We were sorry for our careless mistakes. Thank you for your reminder. In fact, 'OQs'

are the 'MG4s' mentioned in this paper, and we have corrected them in the revised manuscript.

7) Line 137-140, the algorithm used for PG4 definition should be mentioned (Quadparser).

[Reply]: Based on your suggestions, we have given the definition of PG4 in the method section as follows:

PG4 identification and region classification »
PG4s were predicted from hg19 using g4predict (<https://github.com/mparker2/g4predict.git>) according to the Quadparser algorithm by searching for the '(G{3,}[ATCG]{1,7}){3}G{3,}' pattern. For parameter determining step, the start site of 'positive region' is defined as the sequence up to 12 bases (approximate footprint of DNA polymerase) up-stream of the PG4 start site, and the end site was 75 bases down-stream of the start site. After removing the positive regions and 300 bases upstream and downstream of them, the remaining region was split into 75-base windows, which were defined as 'negative regions'. »

Reviewers' Comments:

Reviewer #1:

None

Reviewer #2:

Remarks to the Author:

The authors have responded to my comments. As stated in the initial review, this manuscript proposes a transferable technology for genome-wide G4 detection, which is in need since the actual G4-seq method cannot be used by most (if not all) laboratories. Authors should ensure that all generated scripts are made available with accession codes provided. As presented in the manuscript, methods are detailed but codes for G4-miner should be available (eg. low-quality region scanner). In conclusion, I recommend this article for a publication in Nature communications.

Reviewer: 2

Comments:

The authors have responded to my comments. As stated in the initial review, this manuscript proposes a transferable technology for genome-wide G4 detection, which is in need since the actual G4-seq method cannot be used by most (if not all) laboratories. Authors should ensure that all generated scripts are made available with accession codes provided. As presented in the manuscript, methods are detailed but codes for G4-miner should be available (eg. low-quality region scanner). In conclusion, I recommend this article for a publication in Nature communications.

[Reply]: Thank you for your valuable suggestions to improve the quality of our manuscript. All generated scripts have been deposited in the Github repository (<https://github.com/tulabcode/G4-Miner>, doi: 10.5281/zenodo.5516275).